# The influence of *Pseudomonas syringae* on water freezing and ice melting

**Maria A. Majorina, Victoria R. Veselova, Bogdan S. Melnik**⊙ *

Institute of Protein Research, RAS, Pushchino, Moscow Region, Russia

* bmelnik@phys.protres.ru

## Abstract

*Pseudomonas syringae* is a widely spread plant pathogen known to have ice-nucleating proteins that serve as crystallization sites promoting ice growth at near-zero temperatures. Three temperatures that characterize water freezing and ice melting are (i) the freezing point of water, (ii) the temperature of coexistence of ice and water, and (iii) the melting point of ice. Here we show the influence of different concentrations of *P. syringae* on these three parameters. *P. syringae* appears to affect both the freezing point of water and the temperature of the coexistence of ice and water. Additionally, we propose a research technique for studying the freezing/melting process that is simple and requires no complex equipment.

## Introduction

The ability of bacteria to initiate water freezing at near-zero temperatures was first discovered in 1974 using *Pseudomonas syringae* [1] and attributed to ice-nucleating proteins (INPs) attached to the bacterium's outer membrane. INPs serve as nuclei of water crystallizing [2, 3]. The process of water freezing itself is quite complex, but its main aspects that are relevant to our work are discussed in article "Some peculiarities of water freezing at small sub-zero temperatures" [4].

The effect of different concentrations of *P. syringae* on water freezing has been the subject of many studies. Mostly, this process was controlled visually [5]. Aqueous or phosphate buffer was supplemented with a range of *P. syringae* concentrations, then droplets of the suspension were applied onto an aluminum plate, cooled from 0 to -10˚C, and incubated for a few minutes, followed by visual counting of the frozen droplets. The temperature of solidifying of a certain number of droplets (e.g., 90%) was usually taken as a quantitative parameter of the process. This or similar methods are reported in [1, 6–8]. In [9], the same method was used to study the effect of lyophilized *P. syringae* cells on $H_2O$ and $D_2O$ freezing. In [10], individual droplets of a *P. syringae* solution were visually monitored and the freezing point of each droplet was recorded. In [11], freezing of *P. syringae* solutions was studied in microtubules under visual control. The above and other studies basically describe the temperature dependence of ice growth initiation at a certain concentration of living or lyophilized *P. syringae*.

The methods of visual droplet monitoring are clear but very labor-consuming and insufficiently accurate.

**Data Availability Statement:** All relevant data are within the article.

**Funding:** All the authors of this work was funded by Russian Science Foundation, grant number 21-14-00268. The funders had no role in study design,

data collection and analysis, decision to publish, or preparation of the manuscript.

**Competing interests:** The authors have declared that no competing interests exist.

In our study, continuous temperature monitoring of a *P. syringae* suspension was used during both cooling and heating. This revealed the effect of different concentrations of *P. syringae* on three parameters: the freezing point of water ($T_f$), the temperature at which ice and water coexist in thermodynamic equilibrium ($T_{i-w}$), and the melting point of ice ($T_m$). As a control, the same experiments used different concentrations of *E.coli* cells.

## Results

Fig 1 shows a typical temperature curve for a 0.5 ml water sample during freezing and melting. A test tube with room temperature buffer was placed in a thermostat pre-cooled to -11˚C. A thermometer probe inside the test tube allowed continuous recording of sample temperature. As seen from Fig 1, for about 4 minutes the temperature decreases, with water remaining liquid and supercooled. At some freezing point $T_f$ (in Fig 1, $T_f$ = -9.7˚C), the occurrence of an ice nucleator triggers water crystallization accompanied by heat release. After the 4th minute, the curve goes up dramatically reflecting the increasing temperature. From the 6th to the 9th minute, the sample is a mixture of ice and water that holds the temperature of coexistence of ice and water $T_{i-w}$ (see the plateau in Fig 1, $T_{i-w}$ = -0.11˚C). With the entire water crystalized, the sample temperature decreases towards that of the thermostat (-11˚C).

When transferred to a thermostat pre-heated to +25˚C, the ice sample undergoes heating and melting at around the 16th-17th minute. In Fig 1, this process is seen as a bend in the curve which, ideally, should turn into a horizontal line whose position is determined by the ice melting point ($T_m$). In the case of pure water and an ideal device design, $T_m$ and $T_{i-w}$ must coincide (for details of temperature determination, see Materials and Methods).

Three temperatures that can be derived from the freezing/melting curve of the sample are important for understanding the activities of some proteins. For example, some antifreeze proteins or ice-binding proteins decrease the temperature of coexistence of ice and water $T_{i-w}$. In most publications, this phenomenon is called thermal hysteresis (see, e.g., [12–15]). Other studies show that some proteins can increase the melting point of ice $T_m$ [16, 17]. Using the above method, here we show how different concentrations of *P. syringae* cells with ice-nucleating proteins attached to their outer membrane affect $T_f$, $T_{i-w}$, and $T_m$.

### The effect of cells on water freezing and ice melting

Different concentrations of *P. syringae* cells were used in experiments on water freezing and ice melting, with *E.coli* cells as a control. Fig 2 shows typical freezing and melting curves for samples with different *P. syringae* concentrations. As seen, the presence of *P. syringae* triggers ice formation at higher temperatures, as compared to the buffer. *E. coli* cells do not affect water freezing (Fig 2C), that is, both with and without different *E. coli* concentrations, water freezing occurs at about -10˚C. It should be noted that the freezing process of any solution is associated with the probability of occurrence of an ice nucleator, so the cooling/freezing curves are different even for identical solutions.

In contrast, ice melting is characterized by good reproducibility. Fig 2B shows melting curves for *P. syringae*-containing ice. As seen, the curve shape is virtually unaffected by the difference in *P. syringae* concentration. Similarly, *E. coli* cells in any concentration do not affect the melting point of ice (Fig 2D).

The experiments with *P. syringae* and *E. coli* solutions allowed calculating $T_f$, $T_{i-w}$, and $T_m$ as dependent on cell concentration in the samples. Fig 3 shows these dependencies.

The data shown in Fig 3 allow the following conclusions:

1. None of the parameters $T_f$, $T_{i-w}$, and $T_m$ depends on *E. coli* concentration in the solution.

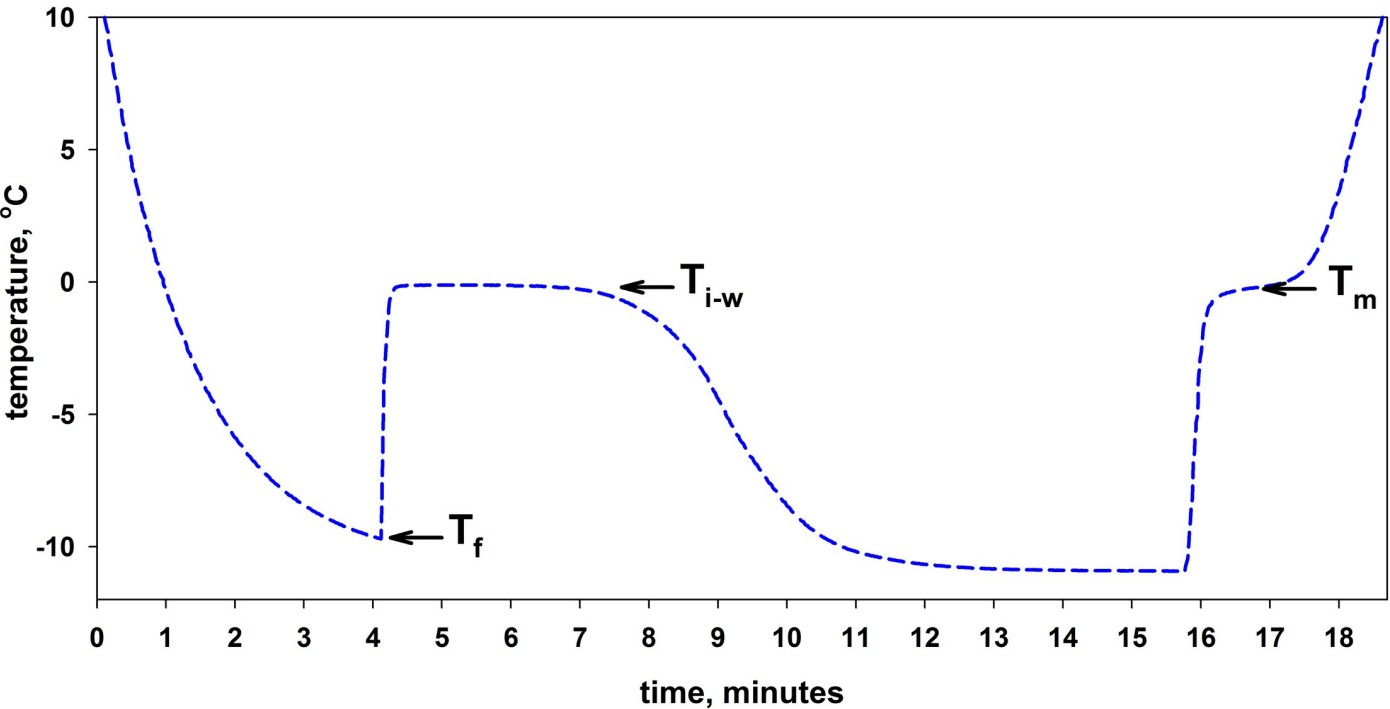

**Fig 1. Time dependence of the sample temperature during cooling and subsequent heating.** A room-temperature sample (buffer, 20mM Tris-HCl, pH 7.5) was kept in a thermostat at -11˚C for the time interval from zero to the 15th minute and after that transferred to a thermostat pre-heated to +25˚C for further monitoring from the 15th minute onwards. The arrows show the curve stretches where the freezing point of water $T_f$, the temperature of coexistence of ice and water $T_{i-w}$, and the melting point of ice $T_m$ were determined. For details of temperature determination, see Materials and Methods.

2. Unlike $T_m$, the parameters $T_f$ and $T_{i-w}$ depend on *P. syringae* concentration in the solution.

Because freezing of supercooled water is a probabilistic process, the results presented in Fig 3A and 3A* are widely scattered, and a large variance is also observed for the buffer $T_f$ data (see Materials and Methods). The probability of ice occurrence depends on the temperature of the solution and the number of ice nucleators. The growing *P. syringae* concentration increases the probability of water crystallizing due to INPs attached to the bacterium's surface. Fig 3A* for *P. syringae* presents a linear dependence that allows calculating the relationship between $T_f$ and the used cell concentrations from the following equation:

$$T_f = 2.5 \log(C) + 0.9, \tag{1}$$

where C is cell concentration measured in optical units (o.u.). A large error (with an average deviation of ±1.5˚C and maximal of 3.4˚C) is explained by the randomness of ice nucleator appearance. This error and the exponential dependence of $T_f$ on cell concentration (Fig 3A) allow the conclusion that at a *P. syringae* concentration above 0.15 o.u., $T_f$ invariably falls within the range from 0˚C to -2˚C. Fig 3A clearly shows that at a cell concentration above 1.5 o.u., the temperatures vary from 0˚C to -2˚C. In other words, it is impractical to use *P. syringae* concentrations above 0.15 o.u. that corresponds to $1.2 \cdot 10^9$ cell/ml.

Fig 3B and 3B* show the dependence of $T_{i-w}$ on *P. syringae* concentration. A relatively small spread of data (± 0.03˚C) results not only from the thermometer accuracy (± 0.01˚C) but rather from a slight difference in ice growth and ice/water distribution observed in each individual test tube. Nevertheless, the dependence of $T_{i-w}$ on *P. syringae* concentration is determined reliably with an average deviation of ±0.03˚C and maximal of 0.05˚C. What can this

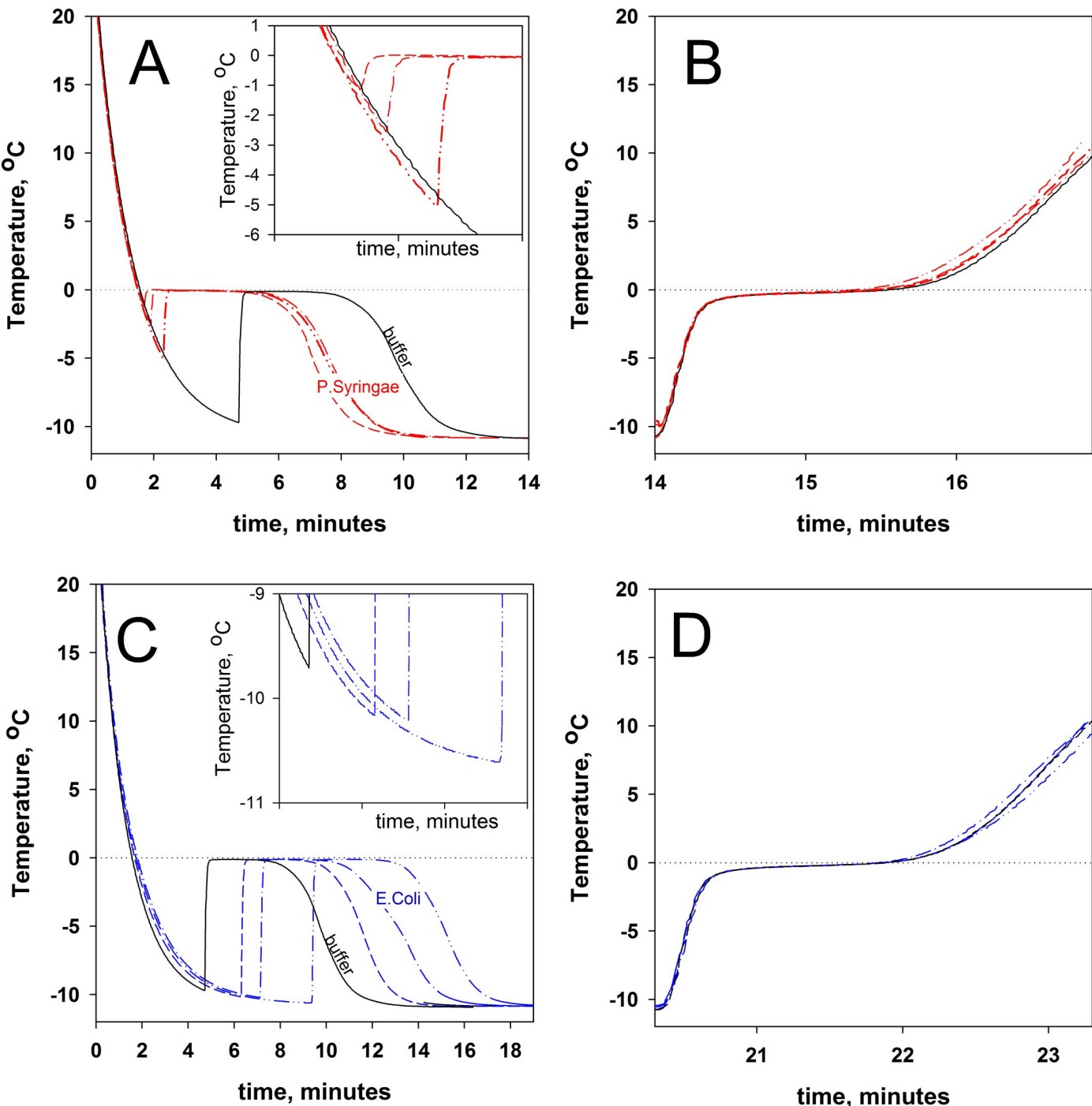

**Fig 2.** Time dependence of sample temperatures during cooling (A and C) and heating (B and D). The curve for the 20mM Tris-HCl buffer, pH 7.5, is shown on all graphs as a black solid line. On A and B, the curves for *P. syringae* solutions with a cell concentration of 0.19 (--), 0.03(···), and 0.003(····) optical units are shown in red. On C and D, the curves for *E coli* solutions with a cell concentration of 0.02(--), 0.2(···), and 0.001(····) optical units are in blue. The inserts on A and C show the curve stretch from which the freezing point of water $T_f$ has been derived.

dependence be related to? Logically, it can be assumed that the cell surface-attached INPs display the ice-binding activity. If the ice surface is stabilized through these interactions, one can expect a shift in the ice melting/freezing equilibrium that will lead to an increase of $T_{i-w}$ with

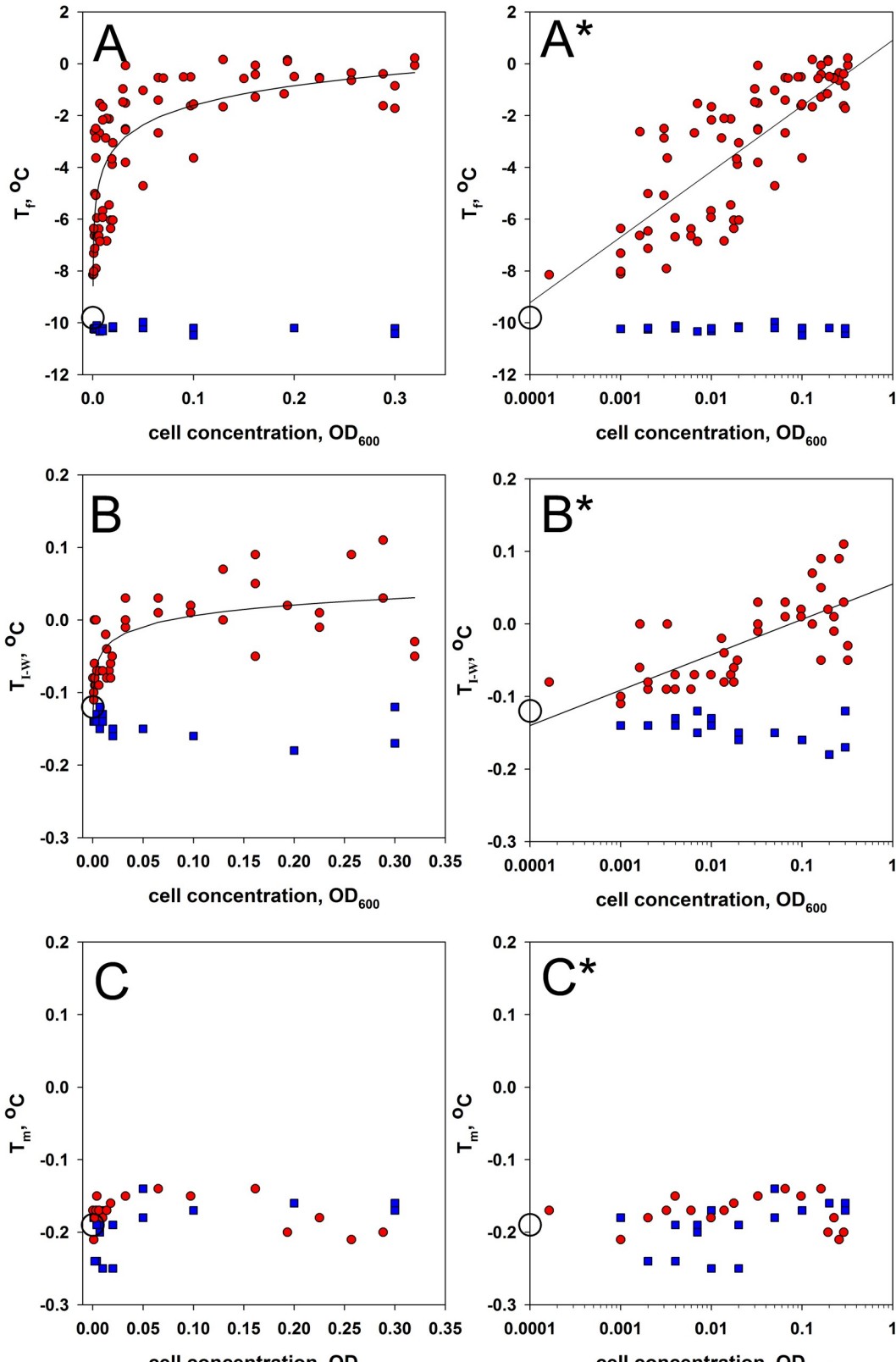

**Fig 3.** Dependence of $T_f$ (A, A*), $T_{i-w}$ (B, B*), and $T_m$ (C, C*) on cell concentration. Figures marked with * are plotted on a logarithmic scale along the concentration axis. Red circles are for *P. syringae* solutions. Blue squares are for *E. coli* solutions. The empty circle shows $T_f$, $T_{i-w}$, or $T_m$ for cell-free buffer averaged across ten replicates.

the increasing number of cells in the solution. This conclusion might seem trivial if not for, e.g., ice-binding proteins whose effect is quite opposite: an increase in their concentration causes a lower $T_{i-w}$ [12–15].

If P. *syringae* cells are assumed to be bound to the ice surface, $T_m$ would also be expected to depend on their concentration. But Fig 3C shows that $T_m$ is totally unaffected by either *P. syringae* or *E. coli* present in the solution. This result is surprising and difficult to explain. It is possible that the reason is the technical peculiarities of the experiment.

When ice grows in supercooled water, the test tube contains a mixture of small ice crystals and water. The thermometer probe is immersed in this mixture, so it measures the average $T_{i-w}$ of the sample accurately enough.

Whereas in the melting experiment, the thermometer probe is surrounded by a large piece of ice whose melting begins from the outer surface close to the test tube walls that are in contact with the thermostat. Therefore, it is not the temperature of a mixture of water and ice that is measured but rather the temperature of the probe-surrounding ice. Perhaps this is the reason why we were unable to detect the dependence of $T_m$ on the *P. syringae* concentration.

## Conclusions

One of the results of this work is the proposed measurement technique. It turned out that using a solid-state thermostat maintaining a certain temperature and a high precision digital thermometer, it is possible to obtain well-reproducible data on peculiarities of water freezing and ice melting. This technique yielded Eq (1) linking the *P. syringae* concentration with the freezing point of a voluminous aqueous solution.

An important result is that this study has first revealed the dependence of $T_{i-w}$ on the *P. syringae* concentration (Fig 3B), which is evidence for *P. syringae* binding to the ice surface. This result might seem trivial since surface INPs of *P. syringae* obviously must interact with the ice surface, but importantly, the obtained dependence means that only free-floating cells interact with ice, unlike those where ice has been already formed.

How to assess whether the interaction of cells with ice is strong or weak? In our experiments, *P. syringae* cells at a concentration of 0.3 o.u. ($\sim 2 \cdot 10^8$ cell/ml) increased $T_{i-w}$ by $\Delta =$ +0.13 $^o$. The equal $T_{i-w}$ decrease ($\Delta = $ -0.13 $^o$) will be observed if the solution is supplemented, e.g., with 70 mM of a low molecular weight substance or 0.25% of ethyl alcohol [18, 19]. Another comparison example is ice-binding proteins which at high concentrations change $T_{i-w}$ by more than 1$^o$ [20]. Thus, despite INPs, the *P. syringae* binding to the ice surface is weak.

## Materials and methods

### Cell culture experiments

*E. coli* BL21 (DE3) cells were grown on LB growth medium (VWR Life Science AMRESCO) at a temperature of 37˚C. *P. syringae* cells (*Pseudomonas syringae pv.syringae*) grown on medium L (yeast extract 5.0 g/l; peptone 15.0 g/l; NaCl 5.0g/l) at a temperature of 26˚C. All cells were grown in a liquid medium up to cell density OD600 = 1.0OU, then precipitated on a centrifuge at 6000g, washed twice with a solution of 20mM Tris-HCl, pH 7.5. The initial cell solution was diluted with a buffer solution of the same composition to the desired concentration. The concentration of cells was controlled by absorption at 600 nm.

### Measurement of water freezing and ice melting

This study used a Biosan CH-100 (Latvia) solid-state thermostat with a fixed temperature of -11˚C to measure water freezing or +25˚C to measure ice melting. 0.5 ml samples in plastic

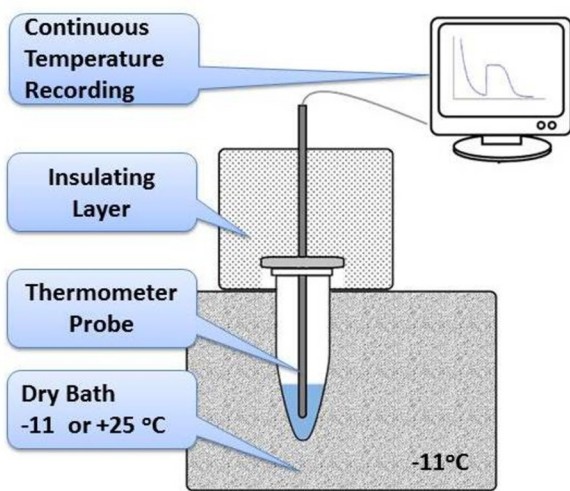

**Fig 4. The device used in the work.** Schematic diagram of a device for water freezing and ice melting studies.

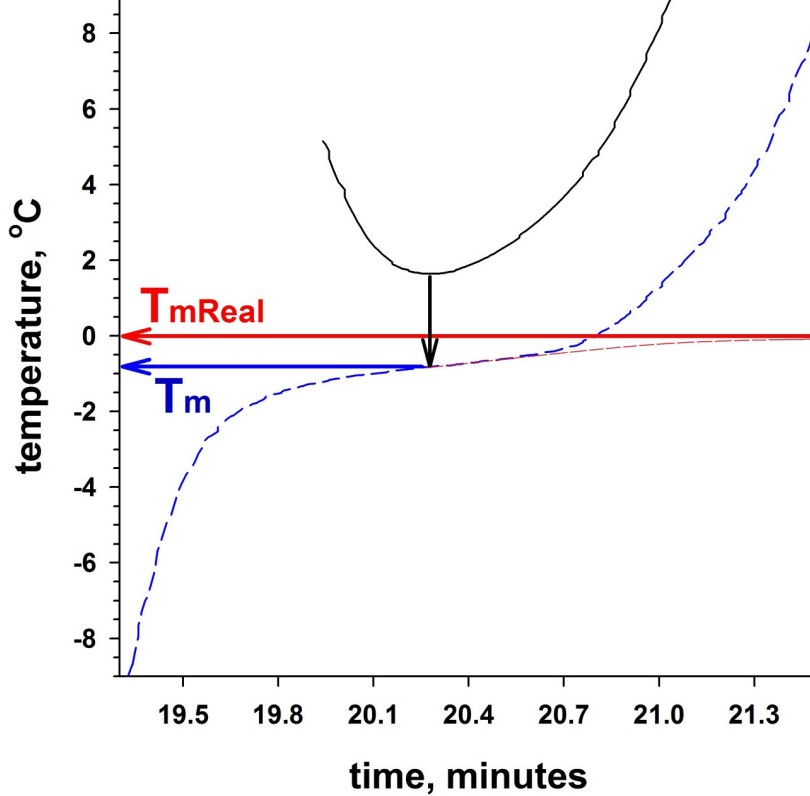

**Fig 5. Determination of the melting point of ice, $T_m$.** The blue dashed line is an example of the time dependence of sample temperature during heating. The black solid line is the first derivative of the blue curve. The minimum of the first derivative graph allows determining the inflection point of the blue dashed curve (shown by the black arrow) from the position of which $T_m$ is determined (shown by the blue arrow). The red dashed line shows a probable change in the experimental curve (blue) for the case when the sample is voluminous and the thermometer can precisely measure the average temperature of the sample. The red dashed line asymptotically tends to the real melting point of ice, $T_{mReal}$. The red arrow shows the real melting point of ice.

1.7 ml test tubes were placed in the thermostat. Temperatures were measured with a digital thermometer LT-300 (Russia) having an absolute accuracy of ±0.05˚C and a relative accuracy of ± 0.01˚C. The metal probe of the thermometer was fixed in a holder and positioned in the center of the tube. The temperature from the thermometer was recorded in real time on a computer. Fig 4 presents a schematic diagram of the described devise.

## Peculiarities of $T_f$, $T_{i-w}$, and $T_m$ determination

From the graph of the temperature dependence of the sample on time (an example is shown in Fig 1) we determined three temperatures $T_f$, $T_{i-w}$ and $T_m$.

The freezing point $T_f$ was determined as the minimum value of the temperature at the time of the beginning of water crystallization in the sample.

The temperature of coexistence of ice and water Ti-w was defined as the temperature of the horizontal section of the curve after the moment of freezing of water. For example, in Fig 1, such a section of the curve is between the 6th and 10th minute.

To determine the melting temperature of ice $T_m$ a graph of the first derivative of the temperature dependence curve was plotted. In Fig 5, the blue dotted line shows the section of the temperature dependence curve where ice melting occurs. The black solid line shows the graph of the first derivative of this section of the curve. The arrows in Fig 4 show how the temperature $T_m$ is determined. Obviously, with such a definition of $T_m$, this parameter will always differ from the actual melting temperature of the $T_{mReal}$ sample. If the sample had a sufficiently

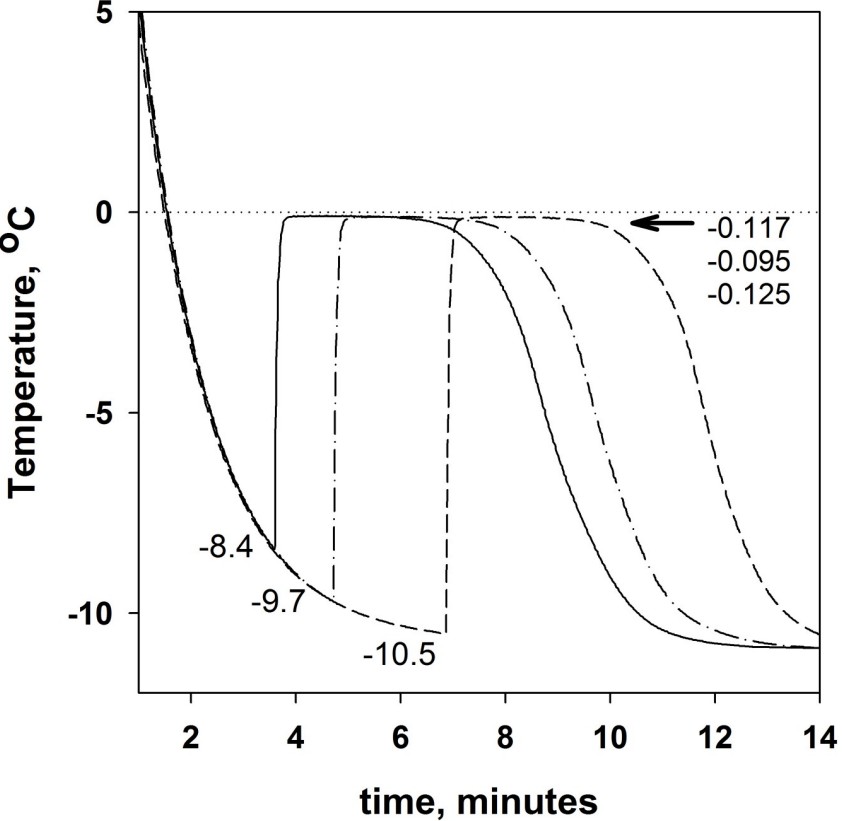

**Fig 6. An example of the time dependence of buffer (20mM Tris-HCl, pH 7.5) temperature during cooling.** For each curve, $T_f$ values are indicated at the bottom of the graph. The $T_{i-w}$ values are indicated at the top of the graph near the arrow. The average values of $T_f$ and $T_{i-w}$ across more than 10 replicates are -9.8 ±0.5° and -0.12±0.01°, respectively.

large volume and the thermometer could correctly measure its average temperature, then the curve would asymptotically tend to the $T_{mReal}$ temperature. In Fig 4, the red dotted line shows the possible shape of the curve with a large sample volume. Moreover, a very accurate determination of the sample temperature is not possible due to the device. The temperature measurement takes place in the middle of the sample (ice), and its melting begins on the walls of the tube (which are in contact with the thermostat). Therefore, the temperature readings are always underestimated (in the case of melting) compared to the average temperature of the sample. Nevertheless, the temperature determination method shown in Fig 5 allows the $T_m$ temperature to be determined in the same way for different samples and to compare the effect of different substances on the melting temperature of ice.

More than 10 measurement replicates of buffer (20mM Tris-HCl, pH 7.5) freezing and melting have been done to obtain the following three parameters: $T_f$ = -9.8±0.5$^o$, $T_{i-w}$ = -0.12 ±0.01$^o$, and $T_m$ = -0.2±0.01$^o$. Fig 1 shows only one curve as an example. Fig 6 shows three buffer freezing curves: one of them with a minimum deviation from the average $T_f$ = -9.8˚C and two others with a maximum deviation. The curves for buffer melting are not plotted here because they are identical to the curves shown in Fig 2B and 2C.

## Acknowledgments

We are grateful to Alexei V. Finkelstein for fruitful discussions. We thank E.V. Serebrova for help in manuscript preparation.

## Author Contributions

**Conceptualization:** Bogdan S. Melnik.

**Data curation:** Maria A. Majorina, Bogdan S. Melnik.

**Investigation:** Maria A. Majorina, Victoria R. Veselova.

**Methodology:** Bogdan S. Melnik.

**Validation:** Maria A. Majorina.

**Writing – original draft:** Bogdan S. Melnik.

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
