## [Decision Letter · Decision Letter 0]

20 Dec 2021

PONE-D-21-36407The influence of Pseudomonas syringae on water freezing and ice melting.PLOS ONE

Dear Dr. Melnik,

Thank you for submitting your manuscript to PLOS ONE. After careful consideration, we feel that it has merit but does not fully meet PLOS ONE’s publication criteria as it currently stands. Therefore, we invite you to submit a revised version of the manuscript that addresses the points raised during the review process.

We look forward to receiving your revised manuscript.

Kind regards,

Lei Li

Academic Editor

PLOS ONE

Journal Requirements:

"We are grateful to Alexei V. Finkelstein for fruitful discussions.

This research was funded by Russian Science Foundation, grant number 21-14-00268"

"The funders had no role in study design, data collection and analysis, decision to publish, or preparation of the manuscript"

Reviewers' comments:

Reviewer's Responses to Questions

**Comments to the Author**

1. Is the manuscript technically sound, and do the data support the conclusions?

Reviewer #1: Yes

Reviewer #2: No

2. Has the statistical analysis been performed appropriately and rigorously? 

Reviewer #1: Yes

Reviewer #2: I Don't Know

3. Have the authors made all data underlying the findings in their manuscript fully available?

Reviewer #1: Yes

Reviewer #2: Yes

4. Is the manuscript presented in an intelligible fashion and written in standard English?

Reviewer #1: No

Reviewer #2: No

5. Review Comments to the Author

Reviewer #1: The language level of the manuscript must be increased. There is also a need to explain the nature of the phenomenon under study much better. Why and how the P. syringae affect the freezing point of water and the temperature of the coexistence of ice and water should be answered on more detailed level. The conditions of the liquid/solid phase transition must be more precisely discussed in the context of physical background of the complex freezing-thawing processes: I. Kratochvilova et al. Theoretical and experimental study of antifreeze protein AFP752, trehalose and dimethyl sulfoxide cryoprotection mechanism: correlation with cryopreserved cell viability, RSC Advances 7 (2017) 352-360.

Reviewer #2: These authors have built a simple apparatus to measure three temperatures in the presence of a bacterium (Pseudomonas syringae) that has surface displayed ice nucleation proteins (INPs). These temperatures are i) the freezing point of water; ii) the temperature at which ice and water co-exist; and iii) the melting point of ice. They have measured these parameters for a wide range of bacterial optical densities in 0.5-mL volumes in a home-made apparatus.

However, it is not clear from the paper what has been discovered. For example, the brief (8-line) Abstract states: “P. syringae appears to affect both the freezing point of water and the temperature of the co-existence of ice and water.” But it does tell us the direction and magnitude of the effects and the scientific basis for them. The last sentence of the Abstract states: “…we propose a novel research technique that is simple and requires no complex equipment.” but it does not say what the technique is for, or what it can accomplish towards solving a scientific problem. These points are also not answered in the paper.

The technique described seems to be a crude version of a manually operated differential scanning calorimeter. This is not a useful instrument for measuring the activity of ice-nucleating bacteria on the freezing point of water because ice nucleation is stochastic event. This parameter is typically measured on small sample volumes (nL to uL) in a multiplexed apparatus where 102 droplets are simultaneously studied and the mid-point where 50% of the droplets are frozen is taken as the nucleation temperature. The authors say: “The methods of visual droplet monitoring are clear but very labor-consuming and insufficiently accurate.” This is simply not true because the appearance of the 102 droplets can be simultaneously recorded through a polarizing filter during the freezing process, and the data can be tabulated in minutes on replay of the video. Another rapid method is to record the exotherms of 102 individual droplets and tabulate the freezing process this way.

The relatively large volume (0.5 mL) of the sample is a problem when measuring the melting process as described in the last paragraph prior to the conclusions. Ice remains around the thermometer probe while the ice melts at the sample edges. This prevented measuring the dependence of the melting temperature on the amount of ice nucleating bacteria present, which was one of the objectives of the study. This seems like a cumbersome technique and it effectiveness has not been demonstrated here.

One way to assess the potential of this system is to do a parallel analysis with ice-binding proteins (IBPs). These proteins have well-documented effects on the temperature at which ice and water co-exist, and the melting point of ice. The former is a temperature range for IBPs that is called the thermal hysteresis value and can be as high as 5-6 C for some insect IBPs. The latter is elevated by a fraction of a degree for most IBPs but would be again a good test of the sensitivity of the apparatus.

This manuscript has been inadequately edited and would have benefitted from a peer review by a colleague. There are many issues to address. It talks about the ‘concentration’ of bacteria. This term applies to something in solution, whereas the bacteria are in a suspension. Cell density is a more appropriate term. E. coli needs a space between E. and coli, as does 20 mM. Elsewhere mM is written as mm.

6. PLOS authors have the option to publish the peer review history of their article (what does this mean?). If published, this will include your full peer review and any attached files.

Reviewer #1: No

Reviewer #2: No

---

## [Author Response · Author response to Decision Letter 0]

24 Feb 2022

Reviewer #1: The language level of the manuscript must be increased. There is also a need to explain the nature of the phenomenon under study much better. Why and how the P. syringae affect the freezing point of water and the temperature of the coexistence of ice and water should be answered on more detailed level. The conditions of the liquid/solid phase transition must be more precisely discussed in the context of physical background of the complex freezing-thawing processes: I. Kratochvilova et al. Theoretical and experimental study of antifreeze protein AFP752, trehalose and dimethyl sulfoxide cryoprotection mechanism: correlation with cryopreserved cell viability, RSC Advances 7 (2017) 352-360.

We have made a lot of corrections, which, in our opinion, have significantly improved the English. The main changes are marked in the text. You can see that we have tried to work on almost every sentence.

As for the detailed discussion of the transition of water from the liquid phase to the solid phase. Unfortunately, this is a very complicated process. Just mentioning the process of water crystallization will not help the understanding of the article, and detailed explanations require some mathematical calculations that are not relevant in this article. We have inserted a reference to article AV. Finkelstein “Some particularities of water freezing at small sub-zero temperatures”, which discusses the process of freezing water and the role of nuclei in ice formation.

Reviewer #2: These authors have built a simple apparatus to measure three temperatures in the presence of a bacterium (Pseudomonas syringae) that has surface displayed ice nucleation proteins (INPs). These temperatures are i) the freezing point of water; ii) the temperature at which ice and water co-exist; and iii) the melting point of ice. They have measured these parameters for a wide range of bacterial optical densities in 0.5-mL volumes in a home-made apparatus.

However, it is not clear from the paper what has been discovered. For example, the brief (8-line) Abstract states: “P. syringae appears to affect both the freezing point of water and the temperature of the co-existence of ice and water.” But it does tell us the direction and magnitude of the effects and the scientific basis for them. The last sentence of the Abstract states: “…we propose a novel research technique that is simple and requires no complex equipment.” but it does not say what the technique is for, or what it can accomplish towards solving a scientific problem. These points are also not answered in the paper.

The whole article is devoted to explaining what technique we use can give. In particular, this method made it possible to continuously monitor the freezing/melting process and made it possible to reveal the influence of PSeringae cells on the temperature of the coexistence of ice and water. Our method allows us to study three temperature parameters (on one installation) and, accordingly, compare them with each other.

Reviewer #2 The technique described seems to be a crude version of a manually operated differential scanning calorimeter. 

Our setup does not look like a bad calorimeter, but a good cryosmometer - an instrument that is usually used to study IBP proteins.

Reviewer #2 This is not a useful instrument for measuring the activity of ice-nucleating bacteria on the freezing point of water because ice nucleation is stochastic event. This parameter is typically measured on small sample volumes (nL to uL) in a multiplexed apparatus where 102 droplets are simultaneously studied and the mid-point where 50% of the droplets are frozen is taken as the nucleation temperature. The authors say: “The methods of visual droplet monitoring are clear but very labor-consuming and insufficiently accurate.” This is simply not true because the appearance of the 102 droplets can be simultaneously recorded through a polarizing filter during the freezing process, and the data can be tabulated in minutes on replay of the video. Another rapid method is to record the exotherms of 102 individual droplets and tabulate the freezing process this way.

We discuss all of the above in the introduction and refer to articles that explore droplets. But when examining droplets, it is impossible to investigate temperature parameters with an accuracy of 0.01 °C. In addition, droplets technique does not allow one to determine the temperature of the coexistence of ice and water.

In our work, we measure temperature parameters with high accuracy, and this is what allows us to draw the main conclusions.

Which technique is more difficult or easier is a moot point. Perhaps there are researchers who will decide that a polarizing microscope should be considered a simple technique, and a thermometer that continuously measures temperature is difficult. We think the opposite.

Reviewer #2 The relatively large volume (0.5 mL) of the sample is a problem when measuring the melting process as described in the last paragraph prior to the conclusions. Ice remains around the thermometer probe while the ice melts at the sample edges. This prevented measuring the dependence of the melting temperature on the amount of ice nucleating bacteria present, which was one of the objectives of the study. This seems like a cumbersome technique and it effectiveness has not been demonstrated here.

One way to assess the potential of this system is to do a parallel analysis with ice-binding proteins (IBPs). These proteins have well-documented effects on the temperature at which ice and water co-exist, and the melting point of ice. The former is a temperature range for IBPs that is called the thermal hysteresis value and can be as high as 5-6 C for some insect IBPs. The latter is elevated by a fraction of a degree for most IBPs but would be again a good test of the sensitivity of the apparatus.

We are discussing the problem of measuring the melting of ice. And we are discussing the fact that the point is not in the installation and its accuracy, but in the fact that it is impossible to start the process of ice melting in the entire volume of ice. Melting always starts at the surface. Therefore, the large volume of the sample (0.5ml) allowed us to somehow measure this parameter. Small droplets melt quickly and it is not possible to fix the exit of the curve to a plateau. We have an idea to make these measurements more accurately, but for this it is necessary to modernize our installation or do joint work with a laboratory that has a “simple” technique for measuring solutions in droplets. In this case, their installation will also have to be changed. Simply put, these measurements cannot be taken quickly. 

Regarding the comparison of the results of our experiments with ice binding proteins. That most IBPs are examined using cryo-osmometers. The cryo-osmometer differs from our installation only in the presence of a vibrator, which "starts" the process of water crystallization at relatively high temperatures (-5 oC). We do not need it, because we have studied the influence of cells on the process of ice nucleation. That is, to put it simply, our work is carried out in exactly the same way as most IBP studies. And, therefore, the results of the study can be compared with works in which a cryo-osmometer is used. We used our facility to study IBP proteins The article in print, but, the results are available in BioRxiv (Glukhova K.A., Okulova J.D., Melnik B.S. (2020) Designing and studying a mutant form of the ice-binding protein from Choristoneura fumiferana. BioRxiv. https://doi.org/10.1101/2020.08.31.275651)

Reviewer #2 This manuscript has been inadequately edited and would have benefitted from a peer review by a colleague. There are many issues to address. It talks about the ‘concentration’ of bacteria. This term applies to something in solution, whereas the bacteria are in a suspension. Cell density is a more appropriate term. E. coli needs a space between E. and coli, as does 20 mM. Elsewhere mM is written as mm.

As for typos. You are absolutely right. Minor typos, such as mm instead of mM, greatly interfere with the understanding of the article. We tried to fix it.

---

## [Decision Letter · Decision Letter 1]

7 Mar 2022

The influence of Pseudomonas syringae on water freezing and ice melting.

PONE-D-21-36407R1

Dear Dr. Melnik,

We’re pleased to inform you that your manuscript has been judged scientifically suitable for publication and will be formally accepted for publication once it meets all outstanding technical requirements.

Kind regards,

Lei Li

Academic Editor

PLOS ONE

Additional Editor Comments (optional):

Reviewers' comments:

Reviewer's Responses to Questions

**Comments to the Author**

1. If the authors have adequately addressed your comments raised in a previous round of review and you feel that this manuscript is now acceptable for publication, you may indicate that here to bypass the “Comments to the Author” section, enter your conflict of interest statement in the “Confidential to Editor” section, and submit your "Accept" recommendation.

Reviewer #1: All comments have been addressed

Reviewer #3: All comments have been addressed

2. Is the manuscript technically sound, and do the data support the conclusions?

Reviewer #1: Yes

Reviewer #3: Yes

3. Has the statistical analysis been performed appropriately and rigorously? 

Reviewer #1: No

Reviewer #3: Yes

4. Have the authors made all data underlying the findings in their manuscript fully available?

Reviewer #1: Yes

Reviewer #3: Yes

5. Is the manuscript presented in an intelligible fashion and written in standard English?

Reviewer #1: Yes

Reviewer #3: Yes

6. Review Comments to the Author

Reviewer #1: The manuscript has been adapted to a form suitable for publication. The authors have sufficiently included all my requirements in the revised manuscript.

Reviewer #3: Obviously, the authors have made great efforts to modify the manuscript. This manuscript is highly innovative and should be published.

7. PLOS authors have the option to publish the peer review history of their article (what does this mean?). If published, this will include your full peer review and any attached files.

Reviewer #1: No

Reviewer #3: No

---

## [Editor Report · Acceptance letter]

2 May 2022

PONE-D-21-36407R1 

The influence of *Pseudomonas syringae* on water freezing and ice melting. 

Dear Dr. Melnik:

I'm pleased to inform you that your manuscript has been deemed suitable for publication in PLOS ONE. Congratulations! Your manuscript is now with our production department. 

Kind regards, 

on behalf of

Dr. Lei Li 

Academic Editor

PLOS ONE